# Graphene Bridge for Photocatalytic Hydrogen Evolution with Gold Nanocluster Co-Catalysts

**DOI:** 10.3390/nano12203638

**Published:** 2022-10-17

**Authors:** Hanieh Mousavi, Thomas D. Small, Shailendra K. Sharma, Vladimir B. Golovko, Cameron J. Shearer, Gregory F. Metha

**Affiliations:** 1Department of Chemistry, University of Adelaide, Adelaide, SA 5005, Australia; 2The MacDiarmid Institute for Advanced Materials and Nanotechnology, School of Physical and Chemical Sciences, University of Canterbury, Christchurch 8140, New Zealand

**Keywords:** gold nanocluster, reduced graphene oxide, SrTiO_3_, photocatalysis, hydrogen evolution reaction

## Abstract

Herein, the UV light photocatalytic activity of an Au_101_NC-AlSrTiO_3_-rGO nanocomposite comprising 1 wt% rGO, 0.05 wt% Au_101_(PPh_3_)_21_Cl_5_ (Au_101_NC), and AlSrTiO_3_ evaluated for H_2_ production. The synthesis of Au_101_NC-AlSrTiO_3_-rGO nanocomposite followed two distinct routes: (1) Au_101_NC was first mixed with AlSrTiO_3_ followed by the addition of rGO (Au_101_NC-AlSrTiO_3_:rGO) and (2) Au_101_NC was first mixed with rGO followed by the addition of AlSrTiO_3_ (Au_101_NC-rGO:AlSrTiO_3_). Both prepared samples were annealed in air at 210 °C for 15 min. Inductively coupled plasma mass spectrometry and high-resolution scanning transmission electron microscopy showed that the Au_101_NC adhered almost exclusively to the rGO in the nanocomposite and maintained a size less than 2 nm. Under UV light irradiation, the Au_101_NC-AlSrTiO_3_:rGO nanocomposite produced H_2_ at a rate 12 times greater than Au_101_NC-AlSrTiO_3_ and 64 times greater than AlSrTiO_3_. The enhanced photocatalytic activity is attributed to the small particle size and high loading of Au_101_NC, which is achieved by non-covalent binding to rGO. These results show that significant improvements can be made to AlSrTiO_3_-based photocatalysts that use cluster co-catalysts by the addition of rGO as an electron mediator to achieve high cluster loading and limited agglomeration of the clusters.

## 1. Introduction

Green Hydrogen (H_2_) is an alternative clean and renewable energy source for the global transition toward net-zero carbon emissions [1]. Currently, over 95% of H_2_ is produced from fossil-fuel feedstocks which is a major contributor to carbon dioxide (CO_2_) emissions [2]. To comply with net-zero carbon emissions guidelines, a complete transition is required for the production, transportation, and consumption of energy. Therefore, H_2_ must be generated from renewable sources such as water and energy created from wind, solar, geothermal, tidal, or biomass reforming [3].

Photocatalytic water splitting is one of the most promising approaches for renewable H_2_ production. The basis of a photocatalyst is a light absorbing semiconductor (i.e., SrTiO_3_, TiO_2_, etc.) and co-catalysts (commonly metal nanoparticles) [4,5]. After absorbing light with energy equal or greater than its band gap (E_g_), a photoexcited electron and a hole are generated in its conduction band (CB) and valance band (VB), respectively [6]. The photoexcited electron can drive reduction reactions, such as hydrogen evolution reaction (HER), while the hole can drive oxidation reactions, such as oxygen evolution reaction (OER). The co-catalyst can increase charge transfer and minimize electron–hole recombination, maximise reactant molecule activation, reaction selectivity, and catalyst stability [7].

The perovskite oxide SrTiO_3_ has been extensively used since 1980 for H_2_ production under UV irradiation due to its outstanding properties [8]. The conduction band minimum (CBM) potential and valence band maximum (VBM) potential make it suitable for H_2_/O_2_ evolution [9]. Doping of Al^3+^ as low-valence cations into the Ti^4+^ sites (AlSrTiO_3_) enhances the photocatalytic H_2_ production up to an apparent quantum yield (AQY) of 96% at 365 nm, which is the highest reported value for metal oxide photocatalysts [10,11,12].

Recently, photocatalytic systems incorporating sub-2 nm sized gold nanoclusters (AuNCs) with a specific number of Au atoms stabilized by ligands have triggered great research interest [13,14,15,16,17]. Typically, AuNCs exhibit non-metallic behaviour (HOMO–LUMO gap), as opposed to localized surface plasmon resonance (LSPR) of nanoparticles (i.e., >2 nm). The presence of HOMO–LUMO gaps in ultra-small AuNCs make them similar to narrow bandgap semiconductors which may be beneficial for photocatalysis [18,19]. AuNCs as co-catalysts can be coupled with semiconductors, resulting in charge transfer which prevents electron–hole recombination [20,21,22]. The catalytic activity of AuNCs is dependent on a number of factors, such as cluster size/structure, ligand type, ligand density, type of support, and metal–support interaction [23]. Use of triphenylphosphine (PPh_3_) ligands provides a simple approach to synthesize AuNCs with small cluster size, narrow size distribution, and easy ligand derivatization. These features make PPh_3_-ligated AuNCs unique and different from AuNCs protected by other ligands [23]. As a result, PPh_3_-ligated AuNCs as the key components of functional materials offer opportunities for both fundamental studies and potential applications, including photocatalysis [16,18,24,25,26].

PPh_3_-ligated AuNCs do not adhere strongly to metal oxide surfaces. Metal oxides have been shown to fragment PPh_3_-ligated AuNCs [27]. It has been reported that the deposition of Au_n_(PPh_3_)_m_ (n: 1, 8, 9, 101) on acidic supports occurs according to two pathways, where the cluster–support interaction is affected by Brønsted and/or Lewis acid sites on the support. The interaction of PPh_3_ ligands with Brønsted acid on the supports results in cluster break down by “oxidative fragmentation” [27]. Lewis acid sites on the support results in ligand migration from the Au clusters to the support without fragmentation of the clusters, and their subsequent agglomeration [27].

We have previously reported that the uniform loading of Au_101_NCs onto reduced graphene oxide (rGO) with no aggregation is facilitated via non-covalent interactions through π–π stacking between the phenyl groups of PPh_3_ and rGO [28]. rGO with *sp^2^*-hybridized 2D structure has extraordinary properties, such as high electrical conductivity, large surface area, charge mobility, and chemical stability [29]. Due to its large surface area, rGO can serve as an “encapsulant” and wrap around semiconductor nanoparticles [30]. Furthermore, rGO as an electron acceptor/mediator facilitates transfer of photo-generated electrons through its π network to the active sites caused by high charge mobility and conductivity, which inhibits electron–hole recombination and promotes photocatalytic H_2_ production [31]. In addition, the tendency of AuNCs to aggregate (either during deposition or activation on the support [32,33]), or deactivate (following initial reaction [34] or under light illumination [20]) is inhibited due to the strong metal support interaction between rGO and AuNCs [28]. The aforementioned features make rGO a desirable candidate to integrate with a broad range of nanomaterials to form nanocomposites with improved performances in photocatalysis.

Herein, we compare two methods to synthesize a photocatalyst incorporating Au_101_(PPh_3_)_21_Cl_5_ co-catalyst, rGO electron mediator, and AlSrTiO_3_ semiconductor. The photocatalytic activity of these nanocomposites was evaluated in photocatalytic HER. Our results highlight the importance of rGO in the synthesis of the nanocomposite to both reduce agglomeration and act as an electron mediator.

## 2. Materials and Methods

### 2.1. Reagents and Materials

All chemicals were used as received throughout the study, unless otherwise stated: Natural graphite flakes (Uley, Eyre Peninsula, South Australia), 98% sulfuric acid (H_2_SO_4_, RCI Labscan), 85% phosphoric acid (H_3_PO_4_, Chem-Supply), 70% nitric acid (HNO_3_, Chem-Supply), 32% hydrochloric acid (HCl, RCI Labscan), 30% hydrogen peroxide (H_2_O_2_, Chem-Supply), potassium permanganate (KMnO_4_, Merck), Au_101_(PPh_3_)_21_Cl_5_ was synthesized following the method described by Hutchison and co-workers [35], AlSrTiO_3_ was provided by K. Domen (University of Tokyo and Shinshu University) and was synthesized following the method described by Ham et al. [10], methanol (CH_3_OH, Merk, Analysis Grade), high-purity Milli-Q water (18.2 MΩ cm at 25 °C), gold single component standard ICP (TraceCERT, Merck, 999 mg L^−1^), 68 Component ICP-MS Standard (High Purity Standards, HPS, 10 µg mL^−1^).

### 2.2. Au_101_NC-AlSrTiO_3_-rGO Photocatalyst Preparation

The procedure for fabrication of AlSrTiO_3_ with controlled content of 1 wt% rGO and 0.05 wt% Au via an ex situ method is schematically presented in Figure 1.

#### 2.2.1. Graphene Oxide, rGO, and Au_101_NC-rGO Synthesis

Graphene oxide (GO), rGO, and Au_101_NC-rGO nanocomposite were synthesized following our previously reported method [28]. In brief, GO was synthesized via the improved Hummers’ method and then was reduced by hydrothermal treatment at 190 °C for 12 h in a 500 mL Teflon-lined stainless-steel autoclave. To obtain the 5 mg Au_101_NC-rGO nanocomposite with 5 wt% Au loading in 1.5 mL methanol, the as-obtained Au_101_NC was dispersed in methanol (1 mg mL^−1^). Then, 320 µL Au_101_NC dispersion (corresponding to 0.25 mg non-ligated Au mass) was added dropwise to 830 µL magnetically stirred as-synthesized rGO dispersion in methanol (5.70 mg mL^−1^) under ambient temperature and made up to 1.5 mL with methanol. The mixture was wrapped immediately with aluminium foil followed by mixing using an incubator and orbital shaker (THERMOstar), for 1 h at room temperature (RT) and 700 rpm.

#### 2.2.2. Au_101_NC-AlSrTiO_3_-rGO Synthesis

The synthesis of Au_101_NC-AlSrTiO_3_-rGO with 1 wt% rGO and 0.05 wt% Au (corresponding to 0.0025 mg non-ligated Au mass) was carried out via two different routes, each with 2 steps, as shown in Figure 1 (Hyphen (-) is used to show the two components are first mixed and colon (:) is used to show the component is added last):(1)Au_101_NC-AlSrTiO_3_:rGO: To obtain 25 mg Au_101_NC-AlSrTiO_3_:rGO, 16.0 µL Au_101_NC dispersion (1 mg mL^−1^ ligated Au mass) was added dropwise to 25 mg of AlSrTiO_3_ dispersed in 5 mL methanol in a porcelain evaporation dish and homogenized using bath sonication at RT until the solvent was completely evaporated. The as-obtained Au_101_NC-AlSrTiO_3_ was dispersed and homogenized in 5 mL methanol via sonication (2 min). Then, 41.7 µL as-synthesized rGO dispersion (5.70 mg mL^−1^) was added dropwise to the dispersion with bath sonication at RT until the solvent was completely evaporated. The as-obtained Au_101_NC-AlSrTiO_3_:rGO was annealed in air in a muffle furnace (S.E.M Pty. Ltd., Adelaide, Australia) at 210 °C for 15 min.(2)Au_101_NC-rGO:AlSrTiO_3_: To obtain 25 mg Au_101_NC-rGO:AlSrTiO_3_, 0.25 mg (75 µL) of the as-synthesized Au_101_NC-rGO nanocomposite was added dropwise to 25 mg of AlSrTiO_3_ dispersed in 5 mL methanol in a porcelain evaporation dish. The dispersion was homogenized using bath sonication at RT until the solvent had evaporated. The as-obtained Au_101_NC-rGO: AlSrTiO_3_ was annealed in air in a muffle furnace at 210 °C for 15 min.

### 2.3. Characterization

Characterization of obtained materials before and after photocatalysis was performed using: scanning electron microscopy (SEM), high-angle annular diffraction field scanning transmission electron microscopy (HAADF-STEM), bright filed scanning transmission electron microscopy (BF-STEM), inductively coupled plasma mass spectrometry (ICP-MS), and UV–visible diffuse reflectance spectroscopy (UV–Vis DRS).

#### 2.3.1. UV–Visible Diffuse Reflectance Spectroscopy (UV–Vis DRS)

UV–Vis DRS measurements were used to infer the extent of Au_101_NC agglomeration and to confirm that co-catalysts (rGO and Au_101_NC) did not affect the bandgap of the photocatalyst (AlSrTiO_3_). UV–Vis DRS measurements were obtained using a spectrophotometer (Cary 5000 UV–Vis–NIR) fitted with a Praying Mantis Diffuse Reflection accessory (Harrick, DRP-SAP). A PTFE disc was used as reflectance standard. For each measurement, the sample holder was filled with approximately 15 mg of the solid-state sample and the reflectance was measured from 200–800 nm.

#### 2.3.2. Scanning Electron Microscopy (SEM)

The surface morphology, agglomeration state of Au_101_NC, and interaction of rGO sheets with AlSrTiO_3_ were measured using a High-Resolution Field Emission Scanning Electron Microscope equipped with EDX Silicon Drift Detectors (FEI-SEM Quanta 450).

#### 2.3.3. High-Angle Annular Dark-Field Scanning TEM (HAADF-STEM)

Images showing the effect of heating and UV irradiation on the size and distribution of Au_101_NC over AlSrTiO_3_ and rGO along with interaction of Au_101_NC, rGO, and AlSrTiO_3_ with each other were acquired with a FEI Titan Themis STEM operating at 200 keV. The STEM was equipped with a Super-X EDS detector in conjunction with a low-background sample holder to minimize Cu background peaks and maximize X-ray collection efficiency. EDS data were analysed using the Velox™ software from Thermo Fisher Scientific. Samples were prepared by dropping prepared dispersions of as-prepared materials (sonicated for 1 min) onto a 300-mesh copper grid with a lacey carbon support film. The solvent was then allowed to evaporate before placing it into the sample holder for analysis. Image J was employed to measure the size of Au particles (200 particles).

#### 2.3.4. Inductively Coupled Plasma Mass Spectrometry (ICP-MS)

ICP-MS (Agilent 8900x QQQ) was employed to determine the total content of Au in Au_101_NC-rGO, Au_101_NC-rGO:AlSrTiO_3_, Au_101_NC-AlSrTiO_3_, and Au_101_NC-AlSrTiO_3_:rGO by measuring the amount of Au that remained in solution (i.e., not adsorbed onto the solid). The as-synthesized sample dispersions in methanol (1 mL) were centrifuged to precipitate any solid, followed by filtration of the supernatant liquid using a Whatman 13 mm, 0.1 µm disposable nylon syringe filter. Then, 0.05 mL of filtrate was taken, and the solvent was allowed to evaporate. To dissolve the remaining solid, 0.2 mL of fresh aqua regia (analysis grade reagents of 32% hydrochloric acid and 70% nitric acid) was added and left for 30 min, then filled up to 25 mL with water for analysis. Gold and phosphorous single standard solutions in 2% *aqua regia* with concentrations of 5, 10, 25, 50, 100 and 200 ppb were used for calibration.

### 2.4. Photocatalysis

Photocatalytic HER was performed in a sealed overhead-irradiation type glass batch reactor (1.7 cm^2^). For each photocatalytic reaction, 7 mg of sample was immersed in 3 mL of methanol:H_2_O (1:2) and sonicated for 1 min to form a homogeneous suspension. Before each reaction, air was evacuated from the system and replaced with Ar (1 atm). The suspension was then irradiated with a UV LED (365 nm, 83 mW/cm^2^, Hongkong UVET Co., UH-82F+L12) for 2 h and stirred during irradiation using a magnetic bar. Starting at 0 h, the evolved gases were sampled hourly and analysed by gas chromatography (Agilent Technologies, 7890B, thermal conductivity detector, Ar carrier gas, molecular sieve 5 Å column).

## 3. Results and Discussion

The 3 component (Au_101_NC, AlSrTiO_3_, and rGO) photocatalysts with 1 wt% rGO and 0.05 wt% Au was prepared by a sequential mixing process via 2 routes. (1) Au_101_NC was first mixed with AlSrTiO_3_ followed by the addition of rGO (Au_101_NC-AlSrTiO_3_:rGO) and (2) Au_101_NC was first mixed with rGO followed by the addition of AlSrTiO_3_ (Au_101_NC-rGO:AlSrTiO_3_) (see Figure 1).

Characterization of the Au_101_NC-AlSrTiO_3_-rGO nanocomposites was performed to investigate the morphology/structural features, agglomeration state, and interactions between the Au_101_NCs, AlSrTiO_3_ and rGO along with their effect on H_2_ evolution.

### 3.1. Physical Characterization of Au_101_NC-AlSrTiO_3_-rGO

Figure 2a,b shows the SEM images of Au_101_NC-AlSrTiO_3_:rGO and Au_101_NC-rGO:AlSrTiO_3_. The SEM images of Au_101_NC-AlSrTiO_3_:rGO (Figure 2a) and Au_101_NC-rGO:AlSrTiO_3_ (Figure 2b) are similar with no obvious change in the morphology/structure and size relative to AlSrTiO_3_ (Appendix A). Although the rGO loading on AlSrTiO_3_ is low (1 wt%) in both nanocomposites, it can be seen in both higher resolution images that the rGO is well dispersed and wraps around some of AlSrTiO_3_ particles.

Figure 3a–f shows the BF-STEM images and log-normal size-distribution histograms for Au_101_NC-AlSrTiO_3_:rGO (Figure 3a–c) and Au_101_NC-rGO:AlSrTiO_3_ (Figure 3d–f). Interestingly, compared with the original cluster (mode (Mo) = 1.52 nm, Appendix A) the cluster size decreases for Au_101_NC-AlSrTiO_3_:rGO (Mo = 1.27 nm) but increases for Au_101_NC-rGO:AlSrTiO_3_ (Mo = 1.73 nm). This suggests the clusters may fragment upon initial interaction with AlSrTiO_3_. This will be further addressed when discussing the ICP-MS results (*vide infra*).

Beyond determining cluster size, imaging was used to locate where the AuNCs were located, and to determine if this differed for the two synthesis routes (e.g., on rGO or AlSrTiO_3_). We had expected to observe more Au_101_NC on the metal oxide in the Au_101_NC-AlSrTiO_3_:rGO sample, considering that the first step is the direct mixing of Au_101_NC with AlSrTiO_3_ (see Appendix A for additional images). The STEM images confirm that regardless of the synthetic process, Au_101_NC was found almost exclusively on the rGO with only very few Au_101_NC (<5) observed on AlSrTiO_3_ amongst the thousands of Au_101_NC observed on rGO within the composite (see Appendix A for example). This observation shows that the PPh_3_ ligated Au_101_NC has a high tendency to interact with rGO, as found in our previous work [28].

For further investigation, ICP-MS was used to determine the amount of Au deposited onto the rGO or AlSrTiO_3_ at each step of the synthetic process. This was achieved by measuring the Au that had not adsorbed (i.e., remained in solution after centrifugation). Table 1 shows the values obtained for the Au adsorption and the consequent Au loading at each step.

When following synthesis route (1); after mixing Au_101_NC and AlSrTiO_3_ together we find that 69% of the Au has adsorbed onto the metal oxide surface. In the case when rGO (in methanol) is added to the Au_101_NC-AlSrTiO_3_, the adsorption increases to 82%. These adsorptions reflect the higher affinity of Au_101_NC for rGO. When following the alternate synthesis route (2); we find an initial high adsorption of 95% (Au_101_NC-rGO) which then decreases to 79% after the addition of AlSrTiO_3_ in methanol. This is likely due to the dissolution of anchored Au_101_NC in methanol (Appendix A). The ICPMS results are consistent with the STEM observation that Au_101_NC are decorating the rGO in the composites with minimal loading on the AlSrTiO_3_ nanoparticles.

### 3.2. Photocatalytic Hydrogen Evolution Activity of AuNCs-AlSrTiO_3_-rGO

Figure 4 presents the effect of co-loading of rGO and Au_101_NC onto AlSrTiO_3_ via different routes on photocatalytic HER performance under UV light irradiation for 2 h under sacrificial conditions (methanol as hole scavenger). For comparison, the photocatalytic activity of rGO, AlSrTiO_3_, Au_101_NC-rGO, Au_101_NC-AlSrTiO_3_, and rGO-AlSrTiO_3_ as control samples under the same experimental conditions were also evaluated. The trend of H_2_ production rate (Table 2) follows: Au_101_NC-AlSrTiO_3_:rGO > Au_101_NC-rGO:AlSrTiO_3_ > rGO-AlSrTiO_3_ ~ Au_101_NC-AlSrTiO_3_ > AlSrTiO_3_ > rGO > Au_101_NC-rGO. The nanocomposite materials Au_101_NC-AlSrTiO_3_:rGO and Au_101_NC-rGO:AlSrTiO_3_ demonstrate the highest activity, producing 385 ± 22 and 334 ± 24 nmol h^−1^ of H_2_, respectively. This is approximately 10 times that of the activity of the rGO-AlSrTiO_3_ and Au_101_NC-AlSrTiO_3_ where AlSrTiO_3_ was only decorated with either rGO or Au_101_NCs. Negative control samples rGO, AlSrTiO_3_, and Au_101_NC-rGO produced negligible amounts of H_2_.

Extended experiments on the system that we did not completely optimize include the effect of Au_101_NC loading. Appendix A indicates that increasing Au mass loading by 20 times (0.05% to 1% wt%) in Au_101_NC-AlSrTiO_3_:rGO results in only a small increase in photocatalytic activity (1.6 times). We also altered the annealing condition of the photocatalyst prior to hydrogen evolution. Appendix A shows that the photocatalytic activity also increases in Au_101_NC-AlSrTiO_3_:rGO in order; air annealing > vacuum annealing > not annealed. Further work is required to ascertain the precise reason why annealing in air produced a more active photocatalyst.

STEM was also undertaken after photocatalysis to observe changes in the Au_101_NC size. Figure 5a–f shows that the average cluster size slightly increased in Au_101_NC-AlSrTiO_3_:rGO-HER (Mo from 1.27 to 1.49), and with almost no change for Au_101_NC-rGO:AlSrTiO_3_-HER (Mo from 1.73 to 1.72). The changes to the mode reflect changes to the most common cluster size. When comparing the mean average and clusters greater than 2 nm, we see there has been some agglomeration in both samples. The amount of Au particles greater than 2 nm increased from 28.6% to 38.5% (13.5% > 3 nm) and 45.0% to 52.9% (14.8% > 3 nm) for Au_101_NC-AlSrTiO_3_:rGO-HER and Au_101_NC-rGO:AlSrTiO_3_-HER, respectively (Table 3).

## 4. Discussion

An efficient photocatalyst must possess three critical properties. A semiconductor with a bandgap able to absorb incident light, an electronic structure able to migrate the photo-generated charges to the particle surface, and the redox potential at the surface-active sites must be suitable for the water splitting reactions. Most photocatalysts, AlSrTiO_3_ included, can only evolve H_2_ with a co-catalyst [36]. This explains the very low H_2_ evolved (<7 nmol h^−1^) in the negative control samples which either do not contain a semiconductor (Au_101_NC-rGO, rGO) or co-catalyst (AlSrTiO_3_). When the semiconductor is combined with one of the co-catalyst materials (Au_101_NC-AlSrTiO_3_ and rGO-AlSrTiO_3_) we observe low H_2_ evolution (30–40 nmol h^−1^). rGO itself has been previously shown to have moderate activity as a HER co-catalyst [37]. Our previous work on electrocatalytic HER showed Au_101_NC was the most active of various gold clusters and far more active than rGO alone [38]. When both rGO and Au_101_NC are combined with AlSrTiO_3_, regardless of synthetic route, we observe a significantly enhanced HER activity (330–390 nmol h^−1^). This is attributed to the contributing role of the rGO to facilitate charge transfer from semiconductor to co-catalyst (Figure 6).

The difference in HER activity of the two synthetic routes that were investigated is small but may be due to the difference in Au_101_NC after deposition with Au_101_NC-AlSrTiO_3_:rGO having a smaller size and slightly greater Au loading. Overall, Au_101_NC-AlSrTiO_3_:rGO shows less agglomeration than Au_101_NC-rGO:AlSrTiO_3_, the differences in clusters size after HER may also be explained by the difference in cluster size before HER, as observed by STEM (Figure 5 and Table 3). The smaller, fragmented, Au_101_NC in the Au_101_NC-AlSrTiO_3_:rGO sample seem to be more prone to agglomeration under UV irradiation. It is difficult to conclude that the agglomeration is caused by the UV light or charge transfer (photoreduction) from rGO and AlSrTiO_3_ to Au_101_NCs. Despite small increases in mean average cluster size, both synthetic routes maintain a mode diameter of gold as clusters (<2 nm). This is supported by UV–Vis DRS in which the plasmonic band of the AuNCs was not observe in both of Au_101_NC-AlSrTiO_3_:rGO and Au_101_NC-rGO:AlSrTiO_3_ (Appendix A). This highlights the strong interaction between Au_101_NCs and rGO prevents cluster agglomeration, which is in agreement with our previous work [28].

These results demonstrate that the properties of the support determine the agglomeration/fragmentation state of Au_101_NC and the type of interactions between Au_101_NCs, AlSrTiO_3_, and rGO which has a large impact on the photocatalytic activity. The introduction of rGO with a large surface area provides more sites for the adsorption of Au_101_NCs. The synergistic effect between Au_101_NCs and rGO and higher tendency of Au_101_NC to interact with rGO results in agglomeration resistance and migration of Au_101_NC from AlSrTiO_3_ to rGO. Such selective loading of Au_101_NCs on rGO over AlSrTiO_3_ improves the photogenerated exciton separation. The rGO acts as a charge carrier resulting in improved transfer of photo-generated electrons through its π network to Au_101_NCs as the active site for H_2_ production. Our findings suggest that the proposed methodology using Au_101_NCs with PPh_3_ ligands to obtain atom-specific metal clusters can be used in photocatalysis with reduced agglomeration without adding a protecting overlayer (e.g., Cr_2_O_3_ overlayer for Au_25_ loaded on BaLa_4_Ti_4_O_15_) [39,40].

The majority of studies on Au co-catalyst systems in photocatalysis focus on nanoparticles where the LSPR is used to extend light absorption into the visible [41,42]. Future work may look at synthesizing size-specific AuNCs which are large enough to exhibit LSPR.

Cluster co-catalysts from metals other than gold have been used in photocatalysis. High density and uniform deposition of Fe, Co, and Ni clusters with sizes less than 1 nm on TiO_2_ significantly increased the photocatalytic H_2_ evolution activity caused by efficient carrier separation [43]. The photocatalytic H_2_ production of Pt_n_ cluster (n: 8, 22, 34, 46, 68) cluster is affected by the size (number of atoms) of cluster [44]. Therefore, our developed methodology can also be applied with other size-specific PPh_3_-ligated metal clusters such as Pt, Pd, Cu, Ni, Co, Ag, and Ir to design highly efficient clusters/rGO nanocomposites for photocatalytic H_2_ production.

## 5. Conclusions

The simple preparation of Au_101_NC-AlSrTiO_3_:rGO as a HER photocatalyst is presented for the first time. The incorporation of rGO and Au_101_NC with AlSrTiO_3_ increases the catalytic activity of the Au_101_NC-AlSrTiO_3_:rGO about 64 times with no change in electronic structure and optical properties of the AlSrTiO_3_. This is attributed to the small particles size and high loading of Au_101_NC co-catalysts enabled by addition of rGO to the composite. In addition, the selective loading of Au onto rGO over AlSrTiO_3_ improves electron–hole separation and facilitates fast charge transfer to the active site and promotes photocatalytic H_2_ production. This methodology provides a simple and new avenue to design photocatalysts using graphene-PPh_3_-ligated AuNCs as effective co-catalysts for photocatalytic reactions.

## Figures and Tables

**Figure 1 nanomaterials-12-03638-f001:**
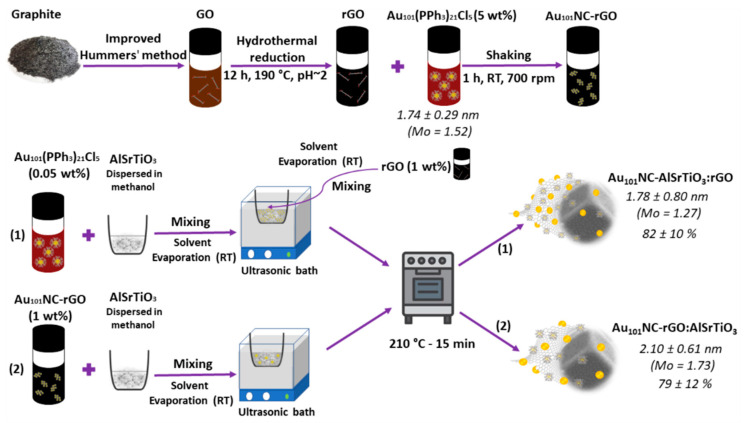
Schematic illustration of synthesis of Au_101_NC-rGO nanocomposite and (1) Au_101_NC-AlSrTiO_3_:rGO and (2) Au_101_NC-rGO:AlSrTiO_3_ with 1 wt% rGO and 0.05 wt% Au. The values on the right-hand side show the Au cluster diameter measured by TEM and final gold loading measured by ICP-MS.

**Figure 2 nanomaterials-12-03638-f002:**
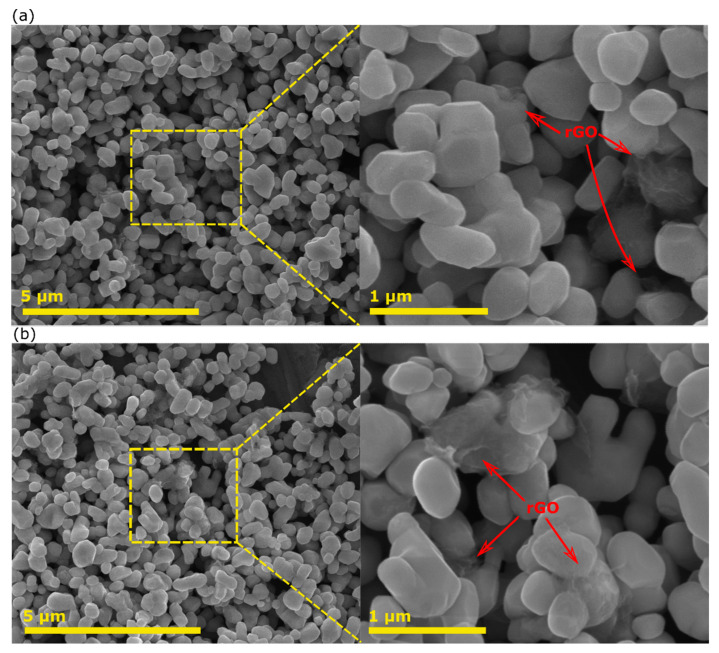
SEM image of (**a**) Au_101_NC-AlSrTiO_3_:rGO and (**b**) Au_101_NC-rGO:AlSrTiO_3_.

**Figure 3 nanomaterials-12-03638-f003:**
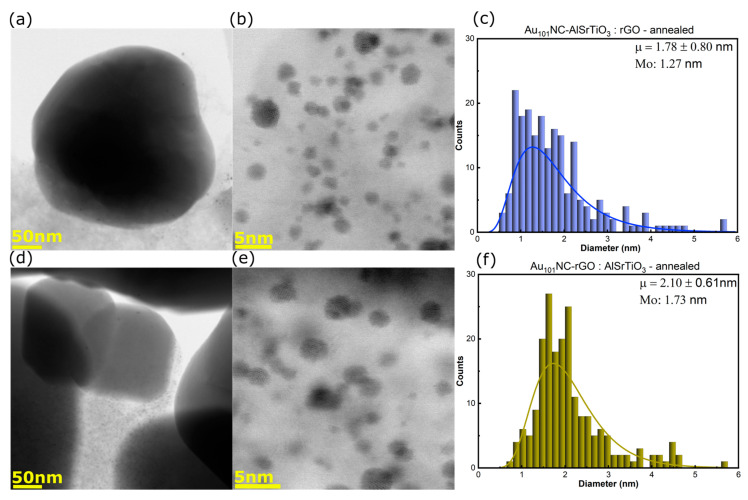
(**a**,**b**) BF-STEM images and (**c**) size distribution histogram of Au_101_NC-AlSrTiO_3_:rGO, (**d**,**e**) BF-STEM images and (**f**) size distribution histogram of Au_101_NC-rGO:AlSrTiO_3_. Histogram fit to log-normal distribution with labels indicating mean (μ ± standard deviation) and Mo. Annealing was performed in air at 210 °C for 15 min. AlSrTiO_3_ and Au_101_NCs are the particles >50 nm and <5 nm, respectively. The grey thin sheets are rGO.

**Figure 4 nanomaterials-12-03638-f004:**
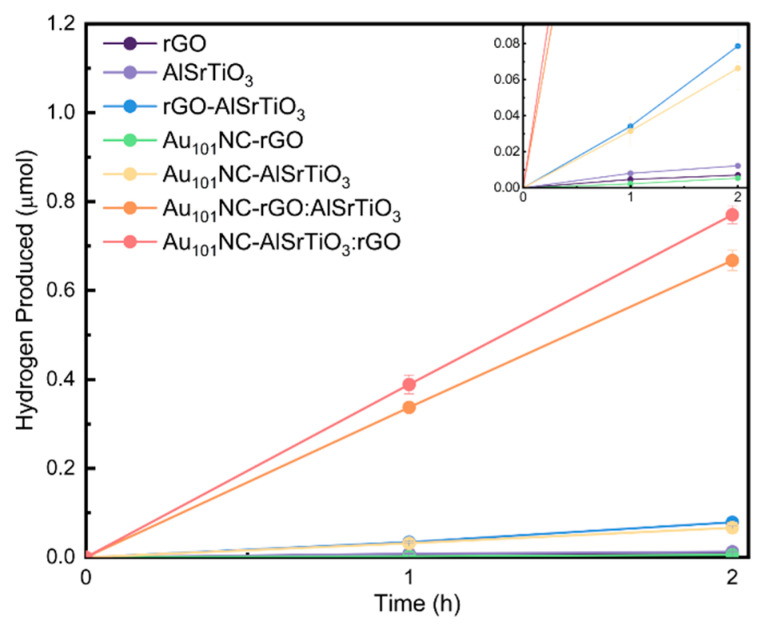
Liquid-phase sacrificial photocatalytic H_2_ production of rGO, AlSrTiO_3_, Au_101_NC-rGO, Au_101_NC-AlSrTiO_3_, rGO-AlSrTiO_3_, Au_101_NC-AlSrTiO_3_:rGO, and Au_101_NC-rGO:AlSrTiO_3_. (Conditions: 1:2 methanol:water, LED 365 nm at 83 mW/cm^2^ for 2 h). Error bars represent standard error. The inset image is scaled up to display the relative activity of rGO, AlSrTiO_3_, Au_101_NC-rGO, and Au_101_NC-AlSrTiO_3_.

**Figure 5 nanomaterials-12-03638-f005:**
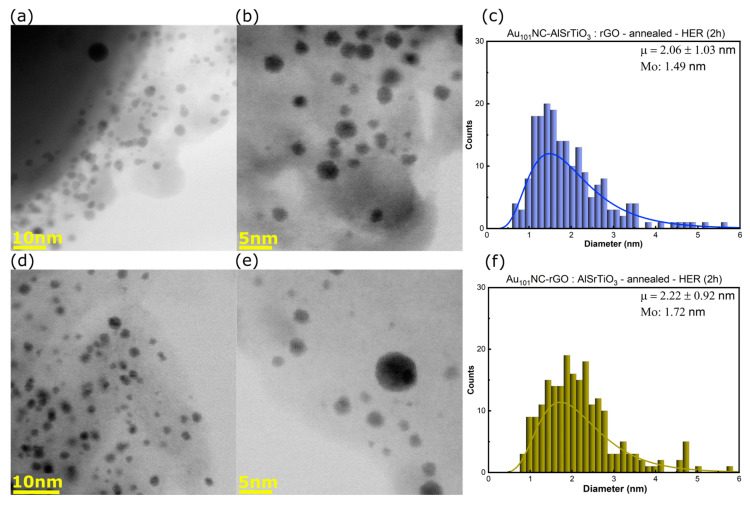
(**a**,**b**) BF-STEM images and (**c**) size distribution histogram of Au_101_NC-AlSrTiO_3_:rGO-HER, (**d**,**e**) BF-STEM images and (**f**) size distribution histogram of Au_101_NC-rGO:AlSrTiO_3_-HER. Histogram fit to log-normal distribution with labels indicating mean (μ ± standard deviation) and mode (Mo). Annealing was performed in air at 210 °C for 15 min. AlSrTiO_3_ and Au_101_NCs are the particles >50 nm and <5 nm, respectively. The grey thin sheets are rGO.

**Figure 6 nanomaterials-12-03638-f006:**
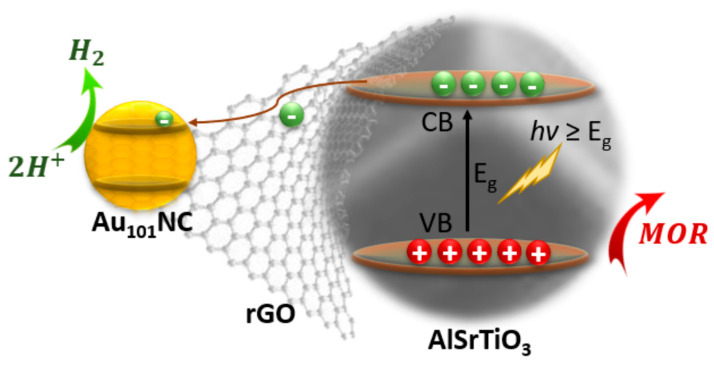
Possible mechanism of electron transfer during photocatalytic H_2_ evolution under UV light irradiation. Electrons are transferred from AlSrTiO_3_ to the CB of Au_101_NC via rGO. (MOR = methanol oxidation reaction).

**Table 1 nanomaterials-12-03638-t001:** Adsorption (%) of Au on AlSrTiO_3_ and rGO by ICP-MS.

	Au Adsorption (%)
Au_101_NC-AlSrTiO_3_	69 ± 11
Au_101_NC-AlSrTiO_3_:rGO	82 ± 10
Au_101_NC-rGO	95 ± 2
Au_101_NC-rGO:AlSrTiO_3_	79 ± 12

**Table 2 nanomaterials-12-03638-t002:** Photocatalytic H_2_ production rate—with Au mass loading 0.05% (0.0025 mg non-ligated Au): 1:2 methanol:water, 365 nm at 83 mW/cm^2^, 2 h reaction time. Error is standard deviation.

Photocatalyst (7 mg)	H_2_ Production Rate (nmol h^−1^)
rGO	4 ± 0
AlSrTiO_3_	6 ± 1.4
Au_101_NC-rGO	3 ± 0
Au_101_NC-rGO:AlSrTiO_3_	334 ± 24
Au_101_NC-AlSrTiO_3_	33 ± 13
Au_101_NC-AlSrTiO_3_:rGO	385 ± 22
rGO-AlSrTiO_3_	39 ± 9.4

**Table 3 nanomaterials-12-03638-t003:** A comparison of particle size in Au_101_NC, Au_101_NC-AlSrTiO_3_:rGO, and Au_101_NC-rGO:AlSrTiO_3_ before and after HER. Annealing was performed in air at 210 °C for 15 min.

	Mean (nm)	Mode (nm)	>2 nm	>3 nm
Au_101_NC	1.74 ± 0.29	1.52	25.6%	3.1%
Au_101_NC-AlSrTiO_3_:rGO	1.78 ± 0.80	1.27	28.6%	9.9%
Au_101_NC-AlSrTiO_3_:rGO-HER	2.06 ± 1.03	1.49	38.5%	13.5%
Au_101_NC-rGO:AlSrTiO_3_	2.10 ± 0.61	1.73	45.0%	12.0%
Au_101_NC-rGO:AlSrTiO_3_-HER	2.22 ± 0.92	1.72	52.9%	14.8%

## Data Availability

The majority of data created during this study are available within this manuscript and its Appendix A. All other data is available upon reasonable request to the corresponding author.

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
