# Peer review of "Graphene Bridge for Photocatalytic Hydrogen Evolution with Gold Nanocluster Co-Catalysts"

_nanomaterials, 2022, doi:10.3390/nano12203638_

Round 1

Reviewer 1 Report

The results presented in the article are original and are of interest to researchers in the field of creating photocatalysts, especially for the photocatalytic hydrogen production.

All procedures of synthesis, characterization and study of photocatalytic activity are described fully and clearly. The figures, data in the tables and Supporting Information are representative and confirm the conclusions.

The only remarks for the attention of the authors.

1.      If the authors believe in 96% of the quantum efficiency for pure, given in the referenced articles, then the quantum efficiency for synthesized photocatalysts should also be determined.

2.      It would be more traditional to carry out the brands of devices in the part of characterization, and not in Supporting Information.

3.      The “Adsorption rate (%)” is not a rate. It is “Adsorption (%)”.

Reviewer 2 Report

In this work the authors describe a nice synthetic procedure to prepare different composites for photocatalytic HER. The samples are well characterised before and after the catalysis, however there are some points that should be addressed before publication:

1.       In the ESI the authors report the DR-spectra of the different samples, but they do not comment on them in the main section. It would give a more complete vision of the system if the authors comment in the main section the fact that they cannot observe the plasmonic effect of the Au NP. This is of paramount importance for photocatalytic experiments. The spectra can be reported, as they are now, in the ESI.

2.       In the main section and in the ESI the authors briefly comment that they have studied different annealing conditions yielding different HER performance, which is very interesting. Could the authors comment on this? Which are the differences expected with the different annealing conditions. It is well known in metal that annealing under oxygenic atmosphere can lead to the formation of oxides. To elucidate the main differences further characterisation on the samples annealed under vacuum or unannealed should be carried out to extract some conclusions.

3.       Regarding the ICP-MS quantification of the Au loading on the composites the authors used a different methodology that the one used for the photocatalytic tested samples. Can this different synthetic route lead to different Au loading?

4.       In the final discussion it would be nice if a comparison with other systems in the literature is included.

5.       As a general comment, the nomenclature of the different samples is a bit confusing for the reader, especially the first time. Perhaps shorter abbreviations should be used in order to facilitate the reading.

Finally, there are some minor issues that should be amended before publication:

p. 1 line 38-39 -> “photoexcited electron and a hole are generated in its valence band (VB) and conduction band (CB), respectively” should be changed by: “photoexcited electron and a hole are generated in its conduction band (CB) and valance band (VB), respectively” Since the hole is generated in the VB and the electron is excited to the conduction band.

p. 10 line 295 -> “the properties of support” should be “the properties of the support”

In the references

1.       check the chemical formulas written in all of them since no capital letter or subscripts are in them. For example, reference 6: srtio3 should be SrTiO3

2.       Some of the journal names are written as abbreviations some other with the complete name. It would be nice if they were all cited using the same style.  

Reviewer 3 Report

       The authors describe two methods for the synthesis of rGO-bridged Au101NC-AlSrTiO3 photocatalysts. The physiochemical properties of as-prepared nanocomposites have been comprehensively characterized. Further experiments demonstrated that these nanostructures were capable of photocatalytic water splitting. Overall, this work is very well performed in detailed experimental studies and has implications for photocatalyst design and the study of photocatalytic hydrogen production. Therefore, I recommend the paper to be accepted and published Nanomaterials after addressing the minor issues.

1. Why use AuNC for co-catalyst instead of Pt nanoparticles?

2. The authors did not provide catalyst cycling capability. In addition, the physical and chemical properties of the catalysts before and after cycling should be provided.

3. The AlSrTiO3 and AuNC nanoparticles in Figure 3 and Figure 5 have similar contrast in the BF-STEM image and are not easily distinguishable, the authors need to make a clarification in the figures. In addition, the HADDF-STEM images in S5 and S6 can be moved to the main text.

4. Reference selection is good in the manuscript. It is advisable to add recent articles on general design strategies for photocatalysis. (e.g., 10.3390/nano9030391, 10.1016/j.apcatb.2017.03.077).

Reviewer 4 Report

The present manuscript discussed the photocatalytic hydrogen production of Au101NC-AlSrTiO3-rGO nanocomposite prepared by two routes. It seems the authors did systematic study on the results and characterization as well. Though it needs some modifications or corrections before it should be accepted by the journal.

1.       The authors mentioned the hydrogen generation values in nmol.h-1 in page no 7. Whereas the units given for figure 4 are in µmol. only. Please make the unit’s uniform all over the results and in the text and the units should be in the universal form.  

2.       As shown in Table 2, the sample Au101NC-rGO showed lower H2 generation activity compared to rGO and AlSrTiO3. What is the reason for this negative impact of NC-rGO. Please justify this results.

3.       The English language is not the efficient, it needs to improve further.

Round 2

Reviewer 2 Report

The authors have partially amended the raised concerns. 

Regarding comment 2. Even though this is a study for future publications perhaps it should be mentioned in the text

In comment 4. I was suggesting comparing also with photocatalytic systems using other metals, how are they compared?

Finally, the references are still missing the subindex in the chemical formulas:

TiO2 should be TiO2

Author Response

Thanks. Response Attached.
